# OpenReview forum: "ROOT DEFENCE STRATEGIES: ENSURING SAFETY OF LLM AT THE DECODER LEVEL"
_ICLR.cc/2025/Conference — ICLR 2025 Conference Withdrawn Submission_

### Official Review · Reviewer_NHoM · 2024-10-17

**Soundness:** 2
**Presentation:** 2
**Contribution:** 2
**Rating:** 3
**Confidence:** 2

**Summary:**

This paper introduces the Root Defense Strategy (RDS), a novel approach aimed at improving the security of Large Language Models (LLMs) at the decoding level. Instead of focusing solely on input (prefill-level defenses) or evaluating the entire response (response-level defenses), RDS assesses and corrects the generation process on a token-by-token basis. This step-by-step method adjusts harmful tokens in real-time, leveraging speculative decoding to maintain inference speed. The authors conduct experiments on five models, demonstrating that RDS effectively reduces compliance with harmful queries while preserving helpfulness and enhancing generation speed.

**Strengths:**

1. The paper presents a new defense mechanism at the decoder level, moving away from traditional input-based or whole-output defenses, offering a more granular and continuous risk assessment.
2. Experimental results demonstrate that RDS is a practical and efficient solution, as it enhances security without compromising speed or utility.

**Weaknesses:**

1. The writing could be improved, as there are inconsistencies between the figures and the content. For example, in Figure 2, the legend does not align with the text, making interpretation more difficult.
2. While the paper evaluates several LLMs, some models are outdated, and incorporating more contemporary models would offer better insights into the generalizability of RDS. Moreover, the evaluation lacks coverage of models with varying sizes, such as the Phi-3 series.
3. The paper provides insufficient detail on the implementation of the training process, particularly regarding how the model learns to detect different harmful token patterns.
4. The paper lacks a theoretical framework or deeper analysis explaining why the token-level corrections proposed by RDS are more effective than existing defense mechanisms.

**Questions:**

1. Could you provide some insights about why the token-level corrections proposed by RDS are more effective than existing defense mechanisms?
2. Is RDS consistent in handling different types of harmful queries? Does RDS show consistent defense capabilities across all categories of harmful queries (e.g., malicious instructions, adversarial prompts)?
3. How does RDS perform in multi-turn conversations?

---

> ### Author Response · Authors · 2024-11-21
> **Response to Reviewer NHoM**
>
> #### W1.
> Thanks for carefully pointing out these issues. We have reviewed and revised the entire manuscript, incorporating the following adjustments: In both the legend and associated text of Figure 2, we have included an explanation regarding $i$ to enhance the overall coherence of the article.
>
> Revised text:
> As refusal outputs are distinguished from compliant outputs at the start, we samples the first few tokens to verify the classifier performance on output. Besides, we additionally sample the last token of the output. Figure \ref{ideal} respectively show the visual results of the first $i$ tokens and and the last token of the outputs. The boundary (the black dashed line) separates harmful queries (red cross) and benign queries (blue ircles), which liiustrates that LLMs can naturally discern the harmfulness of the inputs.
>
> Revised legend in Figure 2:
> Performance of the classifier at the decoding from the $i$-th token and last token of the output. $i$ represents the $i$-th token of the output. The harmful tokens are represented by ''harmful+$i$''. The benign tokens are represented by ''harmless+$i$''. The crosses represent the hidden states of output for harmful queries, while the circles represent the hidden states of output for benign queries. See the visual results from the 4-th token to the 7-th in Appendix E.
>
> #### W2.
>
> We appreciate your feedback. To validate the efficacy of our method, we have expanded the evaluation models employed for the baselines, encompassing various scales and levels of raw security performance from 7B to 13B. The detailed results are presented in Table 1 and Table 2 in our original manuscript, which clearly demonstrates that RDS enhances the security of LLM across diverse sizes while maintaining its helpfulness.
>
> Additionally, we have incorporated a less aligned backbone Mistralx7B in our revised paper. The supplementary experimental results are detailed in the table below. We believe that the supplementary experimental results better demonstrate the superiority of RDS.
>
> Added backbone:
> | Model | Defense | Advbench | Malicious Instruct | Xstest  |
> |-|-|-|-|-|
> |  | No defense | 84 | 73 | 92  |
> |  | safety prompt | 33 | 12 | 52  |
> |  |  Self-Remind | 32 | 11 | 52  |
> | Mistralx7B | DRO | 70 | 68 | 88  |
> |  | Self-Exam | 35 | 10 | 49  |
> |  | SafeDecoding | 23 | 9 | 47  |
> |  | RDS | **13** | **6** | **21**  |
>
> #### W3.
>
> Thank you for your feedback. We will provide more detailed implementation of RDS in the camera-ready version of the text. Next, we will explain how RDS achieves to detect the harmful token step-by-step.
>
> RDS trains classifier within representation space without the need for directly training on the LLMs. As DRO has demonstrated that LLMs are naturally discriminant of input, we validate the classifier's ability on output in section 2. Figure 2 indicates that the benign tokens to harmful queries tend to converge towards the benign side with a smaller offset than harmful tokens. In this way, benign tokens will attain higher scores from the classifier than harmful tokens for harmful queries. In the inference stage, RDS employs the classifier in sampling process to assess the harmfulness of tokens in the candidate list, subsequently adjusting the positions of benign token. By reordering the candidate list token-by-token, RDS improves the defenses ability of LLMs.
>
> #### W4 & Q1.
> Thank you for your feedback. Existing defense mechanisms are typically categorized into two groups: one focuses on defending from the input side through methods like filtering or providing security prompts (Self-Remind and DRO). Jailbreaks aim to devise more sophisticated mechanisms that manipulate models into responding by circumventing protective filters with malicious inputs. The second approach involves defending from the output side, employing techniques such as filtering (Self-Exam and SafeDecoding) and string matching (SafeDecoding). Output-side defenses follow generation-then-evaluation paradigm, impacting the model's inference speed. Besides, both strategies encounter challenges in maintaining the model's utility. In comparison to other baselines, RDS offers distinct advantages:
>
> (1) It enhances its security defense capabilities without compromising the utility of LLMs, as evidenced in Tables 1 and 2. Table 1 demonstrates RDS's security defense strengths. While Table 2 showcases its ability to preserve the utility of LLMs, contrasting with baselines that significantly diminish model utility.
>
> (2) RDS performs corrections token-by-token. We have conducted a deeper analysis in Section 4.5 in our revised paper to prove this point. The case study exemplifies our token-by-token correction methodology, distinct from other baselines employing rejection templates.
>
> Furthermore, we have added deeper analysis on the experimental results in Section 4.3 to explain why the token-level corrections proposed by RDS are more effective than existing defense mechanisms in our revised manuscript.

---

> ### Author Response · Authors · 2024-11-21
> **Response to Reviewer NHoM**
>
> #### Q2.
> Thanks for your insightful question. In Table 1, we have conducted experiments on Malicious Instructions. Following your suggestion, we have added another benchmarks HEx-PHI including different types of harmful queries. The updated results are in the table below, which indicates that RDS is consistent in handling different types of harmful queries.
>
> Added benchmark:
> |  | Defense | Mistralx7B | Vicuna-7B | Vicuna-13B | Qwen2 | Llma3 | Llma2  |
> |-|-|-|-|-|-|-|-|
> |  | No defense | 205 | 89 | 46 | 27 | 5 | 0  |
> |  | safety prompt | 84 | 37 | 14 | 13 | 0 | 0  |
> | HEx-PHI | Self-Remind | 79 | 41 | 11 | 0 | 0 | 0  |
> |  | DRO | 106 | 33 | 3 | 13 | 0 | 0  |
> |  | Self-Exam | 88 | 23 | 5 | 0 | 0 | 0  |
> |  | SafeDecoding | 74 | 21 | 6 | 0 | 0 | 0  |
> |  | RDS | **31** | **16** | **4** | **0** | **0** | **0**  |
>
> #### Q3.
> Thanks for your insightful question. The current defense methodology is assessed solely over a single round of dialogue. Consequently, we extended our research by validating it across publicly benchmarks and prominent LLMs. Theoretically, RDS exhibits the capacity to defense in multi-turn conversations. As elucidated in our paper, RDS constitutes defense from the decoder perspective. Regardless of input alterations or advancements, RDS ensures its output's safety on a token-by-token assessment.

---

> > ### Comment · Reviewer_NHoM · 2024-11-30
> >
> > Thank you for your efforts in the rebuttal. I have read the reviews from the other reviewers and I agree with their assessments. While the revised manuscript does enhance the clarity of the paper, I still believe that further improvements are needed. Therefore, I decided to maintain my score.

---

### Official Review · Reviewer_ZfY1 · 2024-10-26

**Soundness:** 3
**Presentation:** 3
**Contribution:** 3
**Rating:** 6
**Confidence:** 4

**Summary:**

The paper finds that LLMs have the ability to identify whether their output is harmful during the decoding stage, and proposes a Root Defence Strategies (RDS) to select harmless candidate tokens based on the harmfulness score from the LLM. Speculative decoding is introduced to boost the decoding speed. The experiments show RDS achieves the best harmlessness and helpfulness trade-off, with only the "root" security (without further fine-tuning).

**Strengths:**

1. The paper is well written, and clearly expressed.
2. The idea of using LLMs to discriminate the harmfulness of its own decoded tokens is novel and sounds interesting.
3. Table. 2 shows RDS do not increase the refusal rate of LLMs. It is a good merit in practical scenarios.

**Weaknesses:**

1. The results in Table. 1 suggest the previous methods like SafeDecoding and Self-Examination have corrected all of risks on harmfulness benchmarks, the saturation results would reduce the significance of the comparison and make it difficult to show the superiority of RDS. Can authors use more challenging benchmarks or less secure LLMs? It makes the results more distinguishable.
2. RDS should be limited by the ability of base LLMs. If a worst LLM which has low discriminative capability of harmful content, I cannot guarantee the RDS is still better than other baselines.

**Questions:**

1. The performance of the classifier determines the effect of the defense strategy. But I cannot find any numerical indicators of classifier's performance. Is it comparable with LLaMA-Guard?
2. Is there any alternatives of classifier, instead of PCA?

---

> ### Author Response · Authors · 2024-11-21
> **Response to Reviewer ZfY1**
>
> We thank the reviewer for the positive rating of our paper. We also appreciate the reviewer for acknowledging the novelty of this work and all the constructive suggestions. Your positive feedback help us to construct a more complete version of RDS. Below is our specific response to your concern.
>
> #### W1 & W2
> Thanks for your insightful feedback. To emphasize the superiority of RDS, we introduce an added harmful benchmark HEx-PHI which contains 11 types of harmful queries. On this supplementary benchmark, RDS demonstrates superior defense capability compared to other defense methods.
>
> Added benchmark:
> |  | Defense | Mistralx7B | Vicuna-7B | Vicuna-13B | Qwen2 | Llma3 | Llma2  |
> |-|-|-|-|-|-|-|-|
> |  | No defense | 205 | 89 | 46 | 27 | 5 | 0  |
> |  | safety prompt | 84 | 37 | 14 | 13 | 0 | 0  |
> | HEx-PHI | Self-Remind | 79 | 41 | 11 | 0 | 0 | 0  |
> |  | DRO | 106 | 33 | 3 | 13 | 0 | 0  |
> |  | Self-Exam | 88 | 23 | 5 | 0 | 0 | 0  |
> |  | SafeDecoding | 74 | 21 | 6 | 0 | 0 | 0  |
> |  | RDS | **31** | **16** | **4** | **0** | **0** | **0**  |
>
> Furthermore, we introduce a less aligned backbone: Mistralx7B-v1.0. The updated findings are detailed in the following table. RDS continues to exhibit exceptional defensive efficacy.
>
> Added backbone:
> | Model | Defense | Advbench | Malicious Instruct | Xstest  |
> |-|-|-|-|-|
> |  | No defense | 84 | 73 | 92  |
> |  | safety prompt | 33 | 12 | 52  |
> |  |  Self-Remind | 32 | 11 | 52  |
> | Mistralx7B | DRO | 70 | 68 | 88  |
> |  | Self-Exam | 35 | 10 | 49  |
> |  | SafeDecoding | 23 | 9 | 47  |
> |  | RDS | **13** | **6** | **21**  |
>
> As explained by DRO, the capability of LLMs to discern harmful queries is innate and unaffected by their own performance limitations. This can be attributed to the expansive training datasets of LLMs, inherently encompassing data aligned with security protocols. Consequently, even a general LLM demonstrates a discerning ability to identify harmful query. The experiment results on Mistralx7B indicate that RDS is still better than other baselines even on a worst LLM.
>
> #### Q1.
> Thanks for your question. We will provide a detailed performance of the classifier in the camera-ready version of the text. DRO has authenticated the classifier's performance on the input of LLMs. In the original manuscript, we showcase the classifier's performance across all dataset in Figure 9. Notably, Custom constitutes the training data for the classifier, while the remaining datasets are classified as out-of-domain. The illustration in Figure 9 highlights the classifier's consistent ability to classify harmful queries effectively across all datasets.
>
> As for LLaMA-Guard, it is a finetuned model based on Llama2 serving as a sentence-level classifier to assess whether the explicit output responds to the harmful query. Our classifier utilizes the implicit hidden state to ascertain the harmfulness of tokens at token level. In terms of application scope, LLaMA-Guard is designed for evaluating the entire generated output, whereas our classifier provides real-time assessments token-by-token during decoding.
>
> #### Q2.
> Thanks for your question. Our primary focus was to ascertain the capability of the large model in discerning harmful token during decoding. Within this study, We specifically used a basic linear classifier to classify the representation of PCA dimensionality reduction. While employing more sophisticated and efficient classifiers could potentially bolster our defense mechanisms, such a pursuit was not within the scope of our original intent. Furthermore, it is worth noting that intricate classifiers often come at the expense of reduced inference speed.

---

> ### Comment · Reviewer_ZfY1 · 2024-11-26
>
> I still cannot clearly see how the performance of the classifier in Figure 9. Figure 9 only provides a visualized decision boundary without any quantitative results. It will be more convincing if authors can show accuracy or F1 score on Advbench, MaliciousInstruct and Xstest dataset. And I still concern if a LLM has no "root" security (without any alignment), can its implicit hidden state be used to classify harmfulness successfully?
> I find another work [1] which has a similar core idea with RDS, but there is no reference in the paper. What is the main difference?
>
>
> [1] MIL-Decoding: Detoxifying Language Models at Token-Level via Multiple Instance Learning. Xu Zhang, Xiaojun Wan

---

### Official Review · Reviewer_5Hxf · 2024-11-01

**Soundness:** 2
**Presentation:** 1
**Contribution:** 2
**Rating:** 3
**Confidence:** 4

**Summary:**

The paper introduces a jailbreak defense strategy (“Root Defence Strategies”—RDS) that makes use of a classifier applied at each step of generation to the next top k tokens to guide the generation towards only producing safe tokens. The authors integrate their method with speculative decoding strategies to make it faster. Their evaluation results across 5 jailbreak datasets and 5 models show early promise.

**Strengths:**

- LLM safety is a crucial topic, where developing strong defense strategies against jailbreaks can be a potentially highly valuable contribution.

- Addressing safety during the decoding stage is a promising approach for mitigating jailbreaks.

- Using a small-scale classifier during generation might point towards efficient decoding-time defenses.

- I like the authors’ choice of presenting preliminary exploratory experiments that motivate the developed method early in the paper.

**Weaknesses:**

While I find certain aspects of the paper interesting/promising, I believe that currently there are some severe methodological and presentation weaknesses that limit the overall contributions of the paper:

**Writing, clarity, and accuracy**

Although, as mentioned, I appreciate the structure of the paper, I find the writing and presentation very difficult to follow and often erroneous. Certain claims, choices, or statements are also made without clear foundations.

For instance, the sentence starting on line 111 refers to “previous studies”, but does not cite them.

Or on lines 195 to 198, the authors write about their motivating experiments presented in Figure 2 that it “illustrates that the classifier cannot accurately determine whether the output is harmful based solely on the model’s overall decoding…”, but even after multiple careful readings, I do not see how this statement can be derived from the experiment that only looks at certain tokens in the decoding. Perhaps the authors can clarify this point in their next revision. Further, they then use this conclusion to extrapolate onto current defense methods that act on the whole response of the LLM to judge if the response was safe, which I find methodologically questionable, as (i) the experiment was limited to the classifier used by the authors, and (ii) the experiment only covers the case where one looks only at the tokens in certain positions in the response.

Another grieving example is the paragraph under Table 3, which is aimed at conveying a key conclusion, however it is very unclear how the results imply that word-level alignment is the key problem and token-level corrections are the answer.

Mostly in the introduction, but also later on, the paper repeatedly makes the claim that response filtering approaches do not work because they introduce a single point of failure, and doing many classifications, such as in the proposed method, is more favorable. While in general I agree with the intuition, the paper does not seem to follow up on this hypothesis in its evaluation (or at any later point) to back it up with evidence.

The paper contains small errors, both in the use of language, citation formatting (citet vs citep), and technical “typos” (e.g., equation 5 should say argmin if my understanding is correct).

**Evaluation, Technical approach and contribution**

While the overall idea of addressing safety during the decoding stage on a token representation-level is promising, it is closely related to [1] and [2]. While [1] is cited and compared to in the experiments, [2] is omitted both from technical and empirical comparisons. Crucially, the technique of [1] is highly similar to the proposed technique of RDS, which warrants a closer discussion outlining the differences and highlighting in which aspects RDS would mean an improvement over [1]. Currently, I find the novelty introduced by RDS over [1] limited.

Further, certain technical choices are unclearly motivated and unconvincing. I understand that a linear model is (arguably) the fastest choice, however, it might be too weak for general use. To test this, perhaps more complex jailbreaks should have been evaluated where responses are made that may appear benign in the first few tokens (where the presented method seems to have stronger discriminative ability).

It is unclear why the linear classifier is applied on the first k principal components, and not on the whole representation. The choice of k is not tested either.

Crucially, the paper claims to introduce a faster method than competing methods, however, this solely depends on the integration with speculative decoding, as evidenced by the generation time experiment, where the presented method performs consistently as the slowest of all when speculative decoding is not applied. Here, for a fair comparison, speculative decoding should also be applied to the competing methods. Additionally, I believe that integrating the presented method with speculative decoding does not constitute a contribution, as it seems that the speculative decoding technique of choice was applied without any involved adjustment on top of the method.

I find the evaluation metric confusing and atypical. What is the rationale behind setting thresholds over the five samples and reporting only a binary score across the five samples responses? Why is the threshold set to 0.5 for refusal? Why not report just the average across the samples?

I think that evaluations on the Custom dataset should be highlighted and treated more carefully, as this is also the dataset that was used to develop the method and train the classifier.

Finally, my most important doubt over this paper lies in the promise of indiscriminate token-by-token classification. I see two key issues with this approach:

- It is slow even with a fast linear model, as shown even by the experiments of the authors themselves.

- The limitations of the classifier can severely limit the capabilities of the model during generation. This is currently also not evaluated well, as the evaluation solely focuses on jailbreak success and refusal, not evaluating the remaining utility in the models either in the content of the non-refused responses or on unrelated utility benchmarks (e.g., MMLU or TQA).

To summarize, I have certain reservations over the effectiveness of the presented technique and believe that the current evaluation does not explore scenarios that would be specifically difficult for this method to handle well enough.

**References**

[1] Z Xu et al. SafeDecoding: Defending against Jailbreak Attacks via Safety-Aware Decoding. ACL 2024.

[2] A Zou et al. Improving Alignment and Robustness with Circuit Breakers. arXiv 2024.

**Questions:**

See my (implicit) questions in the weaknesses section.

---

> ### Author Response · Authors · 2024-11-21
> **Response to Reviewer 5Hxf**
>
> We are glad that the reviewer finds our paper interesting and our evaluations detailed. Thank you for all the feedback you have provided, which helps our paper to be more complete and perfect. We will respond to each paragraph separately
>
> ###  Writing, clarity, and accuracy
> #### P1 & P2
> We are sorry about the writing problems in our manuscript. We have checked them carefully and revised the errors, including:
>
> Adding refenrences: "For instance, the sentence starting on line 111 refers to “previous studies”, but does not cite them." --> Previous study~\citep{zheng2024prompt} has demonstrated that LLMs can discriminate between different types of prefill and use this ability to enhance safety mechanisms.
>
> Unified citation formatting to citep.
>
> #### P3
> We are sorry for the unclear description. In Figure 2, $i$ represents the $i$-th token generated during decoding. Due to varying output lengths of different queries, we utilize i=last to denote the whole output of the LLM. As depicted in Figure 2, the classifier's classification performance on tokens at distinct positions exhibits instability. For instance, for Qwen2, most benign tokens are correctly classified when $i$=2 and $i$=3. However, the classification becomes ambiguous once more when $i$=last, highlighting the classifier's struggle to accurately classify benign token based solely on the model's overall decoding. Consequently, we deduce that assessing generated token of output within one step is inherently flawed. Furthermore, we observe that benign tokens to harmful queries tend to converge towards the benign side with a smaller offset than harmful tokens. That is to say, for harmful queries, benign tokens attain higher scores from the classifier than harmful tokens, signifying a numerical differentiation rather than relying solely on classification results. We suggest assessing tokens based on the distribution of their classifier values rather than solely relying on the classification results. At its core, the classifier is to predict the harmful score based on the hidden state of token generated in different steps instead of classification.
>
> (i) The choice of utilizing a linear classifier is validated by DRO[3]. In DRO, the representation of harmful and harmless queries can be largely distinguished. The boundary between harmful and harmless queries is easily fitted by logistic regression using queries' harmfulness as labels, which we leverage as the classifier. RDS aims to investigate the potential of LLMs in recognizing harmful output. In our preliminary experiment, we visually demonstrated that for harmful queries, benign tokens attain higher scores from the classifier than harmful tokens.
>
> Figure 2 illustrates that the output feature space does not conform to the previously mentioned boundary. However, we observe that when considering multi-step visualizations, benign outputs for harmful inputs tend to be situated distant to boundary compared to harmful outputs. This is why we use argmax in the equation 5.
>
> (ii) Refusal outputs often start with special tokens, such as "I'm sorry'' or "As an AI''. As refusal outputs are distinguished from compliant outputs at the start, we samples the first few tokens to verify the classifier performance on output. Besides, we additionally sample the last token of the output to represent the whole output.

---

> > ### Author Response · Authors · 2024-11-21
> > **Response to Reviewer 5Hxf**
> >
> > ###  Writing, clarity, and accuracy
> > #### P4
> > Thanks for your feedback. We have added a more in-depth analysis of the results in Table 2 in Section 4.3 in our revised paper.
> >
> > Revised context in Section 4.3:
> > The existing security defense methods have increased the rejection rate of LLMs to a certain extent, especially SafeDecoding and DRO. SafeDecoding utilizes string matching to select the output that corresponds to rejection template. Once the top-$k$ outputs contain the rejection template, SafeDecoding will select the matched one and ignore whether the query is harmful or not. In this way, SafeDecoding significantly compromise the model's utility. This phenomenon is most evident on Xstest, as it contains polysemous words in the query. For instance, ''How to kill time'' is a benign query but contains the harmful word ''kill'' in a conventional sense. On Xstest, the original LLMs intend to reject the query, which increases rejection options in the top-$k$ outputs. Therefore, SafeDecoding tends to reject benign query. DRO moves the representation of harmful/harmless queries in the same/opposite direction of rejection probability, which rely on the initial classification ability of LLMs on input. Figure \ref{xstest} illustrates the classifier's results on all datasets. Notably, LLMs demonstrate robust classification capabilities on all satasets except Xstest. On Xstest, a few of benign inputs are interspersed to the harmful side. This corresponds to the results that the original LLMs is more prone to rejection on Xstest. This misclassification of harmless queries results in DRO aligning the representations of harmless queries with the direction of refusal, resulting in serious rejections on Xstest. In contrast, RDS solely assesses the outputs for defense, disregarding inputs. Thus, the superior utility of RDS on Xstest underscores the benefits of defense mechanisms at the decoder level. Combined with Table 1, it can be included that RDS can enhance the security of LLMs without compromising its helpfulness.
> >
> > #### P5
> >
> > In the experimental section, we conducted pertinent experiments utilizing the Self-Examination and SafeDecoding methods, as detailed in Table 1. Our findings demonstrate that the impact of multiple iterations surpassed that of a single point. Furthermore, we present the subsequent analysis, which serves to provide more robust explaination for your concern.
> >
> > The baselines in our study includes two single-point evaluation methods: Self-Examination and SafeDecoding. Self-Examination assesses the whole output and filters out harmful outputs, essentially evaluating the output based solely on the final step (i=last). On the other hand, SafeDecoding utilizes string matching to select the output that correspond to rejection template, evaluating the output based on the first few steps (i=t, t<last). Despite these methods demonstrating strong security defenses against harmful queries, they significantly compromise the model's utility. As indicated in Table 2, the rejection rate of these baselines for benign queries has also notably increased, suggesting a misjudgment resulting from the single-step filtering approach. By integrating findings from Table 1 and Table 2, RDS amalgamates multiple semantic steps to further bolster the security of LLM without compromising its efficacy.
> >
> > ### Evaluation, Technical approach and contribution
> > #### P1
> > Thank you for sharing the latest work from NIPS2024. We acknowledge that a more thorough comparison with existing literature will bolster the robustness of our paper. In light of the exclusion of reference [2], we intend to include pertinent technical and empirical comparisons in forthcoming versions to ensure comprehensive coverage. In the same way, [2] collects rejection templates. When a rejection template is detected in the output, the power-off module starts to stop the inference of LLMs. Although stopping the generation avoids subsequent unsafe generation, this method, like safedecoding, will increase the rejection probability and reduce the helpfulness of LLMs.
> >
> >
> > #### P2
> > Thanks for your feedback. In the revised paper, we have added a harmful benchmark which contains 11 types of harmful queries. We trust that the extensive benchmarks within different categories can address any uncertainties you may have. The experimental results are listed in the following table.
> >
> > Added benchmark:
> > |  | Defense | Mistralx7B | Vicuna-7B | Vicuna-13B | Qwen2 | Llma3 | Llma2  |
> > |-|-|-|-|-|-|-|-|
> > |  | No defense | 205 | 89 | 46 | 27 | 5 | 0  |
> > |  | safety prompt | 84 | 37 | 14 | 13 | 0 | 0  |
> > | HEx-PHI | Self-Remind | 79 | 41 | 11 | 0 | 0 | 0  |
> > |  | DRO | 106 | 33 | 3 | 13 | 0 | 0  |
> > |  | Self-Exam | 88 | 23 | 5 | 0 | 0 | 0  |
> > |  | SafeDecoding | 74 | 21 | 6 | 0 | 0 | 0  |
> > |  | RDS | **31** | **16** | **4** | **0** | **0** | **0**  |

---

> ### Author Response · Authors · 2024-11-21
> **Response to Reviewer 5Hxf**
>
> #### P3
> Thanks for your question. In this study, we apply PCA to visualize the distribution of hidden state of harmful and harmless output. $k$ represents the reduced dimension of hidden state by PCA. For our experiments, we employed the value as 4 from DRO [3].
> #### P4
> Thanks for your feedback. Speculative decoding is deemed a fundamental element of RDS. During token sampling, the resampling process of the classifier in RDS closely mirrors that of speculative decoding[3][4]. As a result, we integrate this technique into RDS and improve it for our prediction purpose. RDS calculates the harmful score based on the token's hidden state.
>
> The original purpose of speculative decoding is to obtain the hidden state of the candidate token. In LLMs, this hidden state is acquired by multi-layer decoder. The step-by-step generation limit the speed of generation of LLM[5][6][7]. In this work, we replacing the multi-layer decoder with the pre-trained LM head from EAGLE. Our experiments show that this approach significantly accelerates inference while preserving accuracy. It is recognized that our strategy might exhibit reduced efficacy relative to other methods if speculative decoding is omitted. However, current output-side security defense methods operate at the whole output level. For example, SafeDecoding and Self-Examination align with the post-generation evaluation paradigm. Upon detecting harmful output produced by LLMs, a complete output regeneration process is triggered. RDS performs real-time assessments token-by-token during the generation process, ensuring output validity without the need for re-generation. In the future version of our paper, we will demonstrate the time required for inference on harmful benchmarks.
>
>
> #### P5
> Thanks for your feedback. We followed the evaluation method of DRO[3] in our paper. Due to the inherent randomness in LLM's sampling, the outputs may vary with each generation. For each query, we sample five outputs for evaluation. We hypothesize that if more than half of the outputs are rejected to the query, the model is more likely to exhibit a tendency toward rejection. Based on this hypothesize, we set the threshold set to 0.5.
>
>
> #### P6
> Thank you for point out this question. We have removed Custom dataset as the evaluation test. In addition, we select an additional harmful benchmark HEx-PHI for supplementary validation. The experimental results are in the table within our response to P2.
>
> ###
> #### P7
> Thanks for your feedback. In our original manuscript, we have evaluated the non-refused responses of the existing security defense methods and RDS in Table 2. The results prove that RDS matains the helpfulness of LLMs. In Section 4.3, a thorough analysis has been performed to evaluate how various defense methods and RDS affect the efficacy of LLMs. Combined with Table 1, it can be included that RDS can enhance the security of LLMs without compromising its helpfulness. In future versions, we will follow your suggestion to add experimental results on unrelated utility benchmarks.
>
> **References**
>
> > [1] Z Xu et al. SafeDecoding: Defending against Jailbreak Attacks via Safety-Aware Decoding. ACL 2024.
> >
> > [2] A Zou et al. Improving Alignment and Robustness with Circuit Breakers. arXiv 2024.
> >
> > [3] Zhou A, Li B, Wang H. Robust prompt optimization for defending language models against jailbreaking attacks[J]. arXiv preprint arXiv:2401.17263, 2024.
> >
> > [4] Li Y, Wei F, Zhang C, et al. EAGLE: Speculative sampling requires rethinking feature uncertainty, 2024\[J\]. URL https://arxiv. org/abs/2401.15077.
> >
> > [5] Sehoon Kim, Coleman Hooper, Amir Gholami, Zhen Dong, Xiuyu Li, Sheng Shen, Michael WMahoney, and Kurt Keutzer. Squeezellm: Dense-and-sparse quantization. arXiv preprint arXiv:2306.07629, 2023.
> >
> > [6] Heming Xia, Zhe Yang, Qingxiu Dong, Peiyi Wang, Yongqi Li, Tao Ge, Tianyu Liu, Wenjie Li, and Zhifang Sui. Unlocking efficiency in large language model inference: A comprehensive survey of speculative decoding. arXiv preprint arXiv:2401.07851, 2024
> >
> > [7] Tianle Cai, Yuhong Li, Zhengyang Geng, Hongwu Peng, Jason D Lee, Deming Chen, and Tri Dao. Medusa: Simple llm inference acceleration framework with multiple decoding heads. arXiv preprint arXiv:2401.10774, 2024.

---

> ### Comment · Reviewer_5Hxf · 2024-11-21
>
> Thank you for your answers and clarifications.
>
> > Writing
>
> I can still find several grammar errors in the latest pdf, also in newly added passages. I suggest running a grammar checking software like Grammarly on the submission. Communication is important; crude or frequent linguistic errors might diminish the perceived value of even good ideas.
>
> > Clarity and provided motivation from Figure 2
>
> Thank you for the clarification, I also appreciate the update in the paper. However, I am still not convinced with several points here. For instance: "That is to say, for harmful queries, benign tokens attain higher scores from the classifier than harmful tokens, signifying a numerical differentiation rather than relying solely on classification results."---this just means that the optimal decision boundary shifts depending on i. As such, it would be important to discuss how the classifier is trained, and potential improvements to be gained from classifiers that are sensitive to the token position. But mainly I am unconvinced that this token-level experiment's findings extrapolate to the general premise of being able to classify harmful vs. harmless responses (because, if not any time sooner, there is a completion level classification happenning at evaluation time). I do not think that the argument that the last token's hidden state not enabling a clear linear classification of harmful or harmless queries is enough to conclude that responses cannot be classified as harmful or harmless in one step as a whole. Mostly because the final hidden state might carry very little information about the sentiment of the response as a whole, especially if it is a token in a low-entropy context, e.g., being the full stop at the end of the clearly final sentence.
>
> > Speculative decoding
>
> I am still unconvinced about the fairness of the speed evaluations here. There is no fundamental reason for which EAGLE would be more fitting to be applied to RDS than to competing methods. Especially given the fact that the comparison included solely prompting-based defenses.
>
> > Utility evaluations
>
> Non-refusal to a benign query does not imply actual helpfulness. The utility has to be evaluated on utility benchmarks, e.g., as done in [1] and [2].
>
> > Harmfulness metric
>
> Reporting the average is more transparent and in-line with usual practices across most subfields of machine learning.
>
> > Post-rebuttal overall review sentiment
>
> While I appreciate the author's rebuttal, I am firmly confident that this paper has still grieving presentation, methodology, and evaluation limitations that warrant rejection.

---

> > ### Author Response · Authors · 2024-11-22
> > **Response to Reviewer 5Hxf**
> >
> > ### Utility evaluations
> > We acknowledge your clarification. Non-refusal evaluation aims to illustrate that our approach does not bolster security defenses by uniformly increasing the rejection rate of LLMs across all queries. Table 2 demonstrates that existing methods have somewhat raised the overall rejection rate of LLMs. Within Just Eval dataset in [1], utility assessment metrics of LLMs encompass helpfulness, clarity, factual accuracy, depth, and engagement. We argue that evaluating non-refusal scores is akin to assessing the helpfulness of LLMs. If the model outright rejects harmless queries, the outcomes pertaining to these metrics will unquestionably deteriorate.
> > ### Harmfulness metric
> > Following your suggestions, we have revised the. We report the percentages of queries where models generate compliance responses in 5 samplings. Since all datasets, except HEx-PHI, are of size 100, there have been no numerical changes in them. The updated indicators are presented in the table below.
> > Added benchmark: Evaluation results on HEx-PHI. We report the percentages of queries where models generate compliance responses in 5 samplings.
> > |  | Defense | Mistralx7B | Vicuna-7B | Vicuna-13B | Qwen2 | Llma3 | Llma2  |
> > |-|-|-|-|-|-|-|-|
> > |  | No defense | 62 | 89 | 46 | 27 | 5 | 0  |
> > |  | safety prompt | 25 | 37 | 14 | 13 | 0 | 0  |
> > | HEx-PHI | Self-Remind | 24 | 41 | 11 | 0 | 0 | 0  |
> > |  | DRO | 32 | 33 | 3 | 13 | 0 | 0  |
> > |  | Self-Exam | 27 | 23 | 5 | 0 | 0 | 0  |
> > |  | SafeDecoding | 22 | 21 | 6 | 0 | 0 | 0  |
> > |  | RDS | **9** | **16** | **4** | **0** | **0** | **0**  |

---

> ### Author Response · Authors · 2024-11-22
> **Response to Reviewer 5Hxf**
>
> We sincerely appreciate your time and effort in reviewing our response and offering valuable suggestions.
> ### Writing
> Thank you for your suggestion. We have checked the article to ensure the quality of our paper.
> ### Clarity and provided motivation from Figure 2
> Thanks for your feedback. We believe that you may have a misunderstanding of the content in our Section 2. By visualizing the hidden state, DRO identifies a clear boundary between harmful and benign queries within the spatial distribution of representations. This boundary is fitted by logistic regression with the label of queries. Hence, we explore LLMs ability to recognize benign output at the decoder level. Additionally, we investigate the transferability of the classifier from input to output. These aspects have been scrutinized and analyzed in Section 2. Upon analysis in Figure 2, we deduce that benign tokens are closer to the harmful side than harmful ones. Furthermore, this differential distribution can be measured by evaluating the token's value from the classifier that the scores of the benign tokens for hamful queries are larger than that of the harmful tokens. But making a single-step judgment is insufficient to ensure the safety of the following generation. Consequently, we propose multi-steps, token-by-token assessment, consistently opting for the token with the highest classifier score during sampling.
>
> In the field of NLP, the hidden state of the last token is utilized to encapsulate the semantic information of the entire sentence [2][3]. Compared to tokens at other positions, the last token consolidates all the essential information about the sentence. Hence, it is justifiable to employ the last token as a representative of the output. Our assertion is that relying solely on the hidden state of single step (i=t) is unjustifiable, where t represents any step, including the last step. The current step confirms the safety of the immediate output without guaranteeing the safety of subsequent outputs. Consequently, we argue that ''making a single-step judgment is inadequate for determining the harmfulness of the output''.
> ### Speculative decoding
> Thanks for your feedback. The speed of inference in LLMs is constrained by the token-by-token generation. Speculative decoding allows for multiple tokens at one step [4][5][6]. Within the RDS, the LM Head in EAGLE predicts the hidden state of a candicate token based on the previous step's hidden state and the embedding of the candicate token. RDS computes the harmfulness scores of top-k candidate tokens based on the predicted hidden state of them. Like the baseline model, RDS continues to generate tokens sequentially. The primary distinction is in the computation of hidden states. This alteration in the calculation process is deemed a fundamental element of RDS. Hence, we maintain that the speed evaluations is fair.  Moreover, the compared methods are output-based defenses in our prior response. Please consult the baselines introduction in Section 4.1 for more details.
>
> > [1] Z Xu et al. SafeDecoding: Defending against Jailbreak Attacks via Safety-Aware Decoding. ACL 2024.
>
> > [2] Zhou A, Li B, Wang H. Robust prompt optimization for defending language models against jailbreaking attacks[J]. arXiv preprint arXiv:2401.17263, 2024.
>
> > [3] Kenton J D M W C, Toutanova L K. Bert: Pre-training of deep bidirectional transformers for language understanding\[C\]//Proceedings of naacL-HLT. 2019, 1: 2.
>
> > [4] Li Y, Wei F, Zhang C, et al. Eagle: Speculative sampling requires rethinking feature uncertainty[J]. arXiv preprint arXiv:2401.15077, 2024.
>
> > [5] Cai T, Li Y, Geng Z, et al. Medusa: Simple framework for accelerating llm generation with multiple decoding heads[EB/OL].(2023)
>
> > [6] Xia H, Yang Z, Dong Q, et al. Unlocking efficiency in large language model inference: A comprehensive survey of speculative decoding[J]. arXiv preprint arXiv:2401.07851, 2024.

---

> ### Comment · Reviewer_5Hxf · 2024-11-22
>
> > Figure 2 and Section 2
>
> Even if the assertions about the meanings of the token hidden states were true, the statement to be made in Section 2 with the available evidence would only be the following: "With a linear classifier trained on the embeddings of benign and harmful queries, one cannot classify reliably a whole response's safety based on the hidden state of just a single token position." Which is true, but is only motivating why this particular method has to be applied across tokens at each position and not why other methods do not work, which is the claim the authors are trying to make, with which I am disagreeing. Further, the last token hidden state, especially the top one just before the classifier head, does not necessarily compress all the semantic information of the sentence that is relevant for any other downstream task then deciding which given token is the best one to be produced next. Just consider the following example:
> ```
> from transformers import GPT2Tokenizer, GPT2Model
> import torch
> from sklearn.metrics.pairwise import cosine_similarity
> # Initialize GPT-2 tokenizer and model
> tokenizer = GPT2Tokenizer.from_pretrained("gpt2")
> model = GPT2Model.from_pretrained("gpt2")
> # Define two completely different sentences ending with "."
> sentence1 = "The cat sat on the mat."
> sentence2 = "Quantum physics is a fascinating field."
> # Tokenize sentences
> inputs1 = tokenizer(sentence1, return_tensors="pt")
> inputs2 = tokenizer(sentence2, return_tensors="pt")
> # Get hidden states
> with torch.no_grad():
>     outputs1 = model(**inputs1, output_hidden_states=True)
>     outputs2 = model(**inputs2, output_hidden_states=True)
> # Extract the last hidden state of the last token (".")
> last_hidden_state1 = outputs1.hidden_states[-1]
> last_hidden_state2 = outputs2.hidden_states[-1]
> # Get the representation of the last token
> last_token_representation1 = last_hidden_state1[0, -1, :]
> last_token_representation2 = last_hidden_state2[0, -1, :]
> # Compute cosine similarity
> similarity = cosine_similarity(
>     last_token_representation1.unsqueeze(0).cpu().numpy(),
>     last_token_representation2.unsqueeze(0).cpu().numpy()
> )
> print(f"Cosine Similarity between last token representations: {similarity[0][0]:.4f}")
> ```
> Runnig this code produces:
> `Cosine Similarity between last token representations: 0.9967`
>
> Whereas using an actual sentence embedder for instance:
>
> ```
> from sentence_transformers import SentenceTransformer, util
>
> # Load the pre-trained sentence transformer model
> model = SentenceTransformer('all-MiniLM-L6-v2')
>
> # Define the sentences to be compared
> sentence1 = "The cat sat on the mat."
> sentence2 = "Quantum physics is a fascinating field."
>
> # Compute embeddings for the sentences
> embedding1 = model.encode(sentence1, convert_to_tensor=True)
> embedding2 = model.encode(sentence2, convert_to_tensor=True)
>
> # Compute the cosine similarity between the embeddings
> cosine_similarity = util.cos_sim(embedding1, embedding2)
>
> # Print the cosine similarity
> print(f"Cosine Similarity: {cosine_similarity.item():.4f}")
> ```
> Provides the much more meaningful output of: `Cosine Similarity: 0.0095`
>
> > Speculative decoding
>
> I am familiar with how inference in LLMs works and I am familiar with the goals of speculative decoding and how it works. This is why I am confident that EAGLE can be applied to the baselines as well. In the argumentation of the authors, there is nothing that would underline why EAGLE would be only possible to be applied in conjuction with their method and not with other methods, once again, especially together with the prompting baselines.
>
> > Utility evaluations
>
> Yes, if the answer is refused, there is no utility in it. But even a non-refused answer can be a bad answer, this is why competing work executes the mentioned utility evaluations. This is something that is not apriori clear. The very response of the authors underlines the needs for these evals: "Within Just Eval dataset in [1], utility assessment metrics of LLMs encompass helpfulness, clarity, factual accuracy, depth, and engagement.".
>
> > Harmfulness metric
>
> If I understand it correctly, within the 5 samples it is still 1 if "at least once it complied" and 0 if "did not comply in any of the 5 samples". If this is the case, this is not what I was interested in before, I understand that converting the prior score to the reported percentage here is just a trivial scaling. Instead, I am interested in the percentage of the samples for a single query that are refused (assuming non-deterministic sampling) and then these percentages averaged across the queries in the dataset.

---

### Official Review · Reviewer_RW44 · 2024-11-04

**Soundness:** 2
**Presentation:** 2
**Contribution:** 2
**Rating:** 5
**Confidence:** 3

**Summary:**

The paper proposes a defense mechanism. They train a classifier to guide the model to exclude certain unsafe words during decoding. They also integrate speculative decoding the expedite the process.

**Strengths:**

The paper is easy to follow. Discussions are thorough in the paper.

**Weaknesses:**

1. Details regarding the training the classifier are missing in main text. However, this is the main contribution of the paper.

2. What is the advantage of training the classifier compared to simply maintaining a list of unsafe words? Are there direct evaluations on the trained classifier or case study?

3. The results in Table 1 (main results) seem not demonstrate the effectiveness of the method. RDS achieves similar results to several baselines.

**Questions:**

Please refer to the weakness part.

---

> ### Author Response · Authors · 2024-11-21
> **Response to Reviewer RW44**
>
> We greatly appreciate your thoughtful suggestions, which give us a lot of inspiration to revise the paper to make it clearer.
>
> ### W1.
> Thanks for your feedback. We have detailed the training procedure in Appendix C in our original manuscript. Inspired by your feedback, we will put it in the main text in the camera-ready version of text. The classifier equals to a lightweight and effective linear layer. DRO has authenticated that LLMs can recognize the harmful and harmless queries and the classifier can fit to the representations of them. In RDS, our primary innovation lies in validating the classifier's proficiency on the output and leveraging it to execute token-by-token security defense.
>
> ### W2.
> Thanks for your question. The classifier in RDS undergoes training at the context level, offering contextual semantics in contrast to word-level training. As illustrated in Section 4.3, the issue of word ambiguity arises on Xstest. Deeming queries containing a negative word in the conventional sense as harmful ones would be unjustifiable. As the instance illustrated in Section 4.3, ''kill'' is harmful when referring to ''killing a person'' whereas it is benign when referring to ''kill time''. A more informed assessment should be made based on the query in conjunction with its context. Thus, we believe that the context-level classifier in RDS surpasses the word-level classifier in efficacy.
>
> Moreover, we investigate the classification capability on the output. The classifier in RDS can generate safe output token-by-token, constituting a phased accumulation refusal that is more nuanced than directly employing unsafe vocabulary.
>
>
> ### ''Are there direct evaluations on the trained classifier or case study?''
> Thank you for your reminder. We will provide more direct and detailed evaluations on the trained classifier in future versions. In Figure 9 of the original manuscript, we have demonstrated the capability of the classifier on all datasets. Among these datasets, Custom is the training data of the classifier. The others are out-of-domain datasets. As shown in Figure 9, the classifier demonstrates robust classification capability for harmful queries on all datasets.
>
> ### W3.
> Thanks for your feedback. The evaluation of security defenses requires two aspects[1][2]: the security defense capability and the effectiveness of LLMs. If defense strategies solely focus on increasing rejection probabilities, they will significantly reduce the utility of LLMs, making it an unfavorable approach. Tables 1 and 2 demonstrate that the RDS maintains exceptional security defense capabilities while preserving the utility of LLMs. In contrast, alternative baselines, such as Self-Remind[3], showcase commendable security defense capabilities but severely compromise the utility of LLMs (as evidenced in Table 2, LLama2 with a 100% rejection rate). Furthermore, various security defense methods escalate LLM rejection rates to different degrees. In contrast, RDS does not depend on raising LLM rejection probabilities to enhance LLM safety. Tables 1 and 2 collectively emphasize the superiority of RDS.
>
> Additionally, we have incorporated a less aligned backbone Mistralx7B and a benchmark HEx-PHI comprising 11 types of harmful queries. The supplementary experimental results are detailed in the table below. We believe that the supplementary experimental results better demonstrate the superiority of RDS.
>
> Added backbone:
> | Model | Defense | Advbench | Malicious Instruct | Xstest  |
> |-|-|-|-|-|
> |  | No defense | 84 | 73 | 92  |
> |  | safety prompt | 33 | 12 | 52  |
> |  |  Self-Remind | 32 | 11 | 52  |
> | Mistralx7B | DRO | 70 | 68 | 88  |
> |  | Self-Exam | 35 | 10 | 49  |
> |  | SafeDecoding | 23 | 9 | 47  |
> |  | RDS | **13** | **6** | **21**  |
>
>
> Added benchmark:
> |  | Defense | Mistralx7B | Vicuna-7B | Vicuna-13B | Qwen2 | Llma3 | Llma2  |
> |-|-|-|-|-|-|-|-|
> |  | No defense | 205 | 89 | 46 | 27 | 5 | 0  |
> |  | safety prompt | 84 | 37 | 14 | 13 | 0 | 0  |
> | HEx-PHI | Self-Remind | 79 | 41 | 11 | 0 | 0 | 0  |
> |  | DRO | 106 | 33 | 3 | 13 | 0 | 0  |
> |  | Self-Exam | 88 | 23 | 5 | 0 | 0 | 0  |
> |  | SafeDecoding | 74 | 21 | 6 | 0 | 0 | 0  |
> |  | RDS | 31 | 16 | 4 | 0 | 0 | 0  |
>
> **References:**
>
> > [1] Zhou A, Li B, Wang H. Robust prompt optimization for defending language models against jailbreaking attacks[J]. arXiv preprint arXiv:2401.17263, 2024.
>
> > [2] Xu Z, Jiang F, Niu L, et al. Safedecoding: Defending against jailbreak attacks via safety-aware decoding[J]. arXiv preprint arXiv:2402.08983, 2024.
>
> > [3] Xie Y, Yi J, Shao J, et al. Defending chatgpt against jailbreak attack via self-reminders\[J\]. Nature Machine Intelligence, 2023, 5(12): 1486-1496.

---

> > ### Comment · Reviewer_RW44 · 2024-11-29
> >
> > Thank the authors for the responses! I believe including the mentioned details in revisions will help clarity.

---

### Note · Authors · 2024-12-02

**Comment:**

I would like to withdraw my manuscript titled ROOT DEFENCE STRATEGIES: ENSURING SAFETY OF LLM AT THE DECODER LEVEL from further consideration. Thanks for all reviewers' comment.

**Withdrawal Confirmation:**

I have read and agree with the venue's withdrawal policy on behalf of myself and my co-authors.